# Low-Cost 3D Models for Cervical Spine Tumor Removal Training for Neurosurgery Residents

**DOI:** 10.3390/brainsci14060547

**Published:** 2024-05-27

**Authors:** Albert Sufianov, Carlos Salvador Ovalle, Omar Cruz, Javier Contreras, Emir Begagić, Siddarth Kannan, Andreina Rosario Rosario, Gennady Chmutin, Garifullina Nargiza Askatovna, Jesus Lafuente, Jose Soriano Sanchez, Renat Nurmukhametov, Manuel Eduardo Soto García, Nikolay Peev, Mirza Pojskić, Gervith Reyes-Soto, Ismail Bozkurt, Manuel De Jesus Encarnación Ramírez

**Affiliations:** 1Federal State Budgetary Institution the Federal Center of Neurosurgery of the Ministry of Health of the Russian Federation, 625062 Tyumen, Russia; 2Department of Neurosurgery, State Medical University (Sechenov University), 119991 Moscow, Russia; 3Educational and Scientific Institute of Neurosurgery, Peoples’ Friendship University of Russia RUDN University, 117198 Moscow, Russia; 4Department of Neurosurgery, National University of Mexico Hospital General, Durango 34030, Mexico; 5Department of General Medicine, School of Medicine, University of Zenica, 72000 Zenica, Bosnia and Herzegovina; 6School of Medicine, University of Central Lancashire, Preston PR02AG, UK; 7Autonomous University of Santo Domingo (UASD), Santo Domingo 10103, Dominican Republic; 8Petrovsky Russian Scientific Center of Surgery, 121359 Moscow, Russia; 9Spine Center Hospital del Mar, Sagrat Cor University Hospital, 08029 Barcelona, Spain; 10Instituto Soriano de Cirugía de Columna Mínimamente Invasiva at ABC Hospital, Neurological Center, Santa Fe Campus, Mexico City 05100, Mexico; 11NCC No. 2 Federal State Budgetary Scientific Institution Russian Scientific Center Named after. Acad. B.V. Petrovsky (Central Clinical Hospital Russian Academy of Sciences), 121359 Moscow, Russia; 12Department of Neurosurgery, Hospital Angeles Villahermosa, Sante Fe, Mexico City 01210, Mexico; 13Department of Neurosurgery, Russian People’s Friendship University, 117198 Moscow, Russia; 14Department of Neurosurgery, University Hospital Marburg, Baldingerstr., 35033 Marburg, Germany; 15Department of Head and Neck, Unidad de Neurociencias, Instituto Nacional de Cancerología, Mexico City 14080, Mexico; 16Department of Neurosurgery, Medical Park Ankara Hospital, Kent Koop Mah 1868. Sok, Batıkent Blv. No:15, 06680 Ankara, Turkey; 17Department of Neurosurgery, School of Medicine, Yuksek Ihtisas University, 06520 Ankara, Turkey

**Keywords:** cervical spine, laboratory, 3D model training, residents training

## Abstract

Background and Objectives: Spinal surgery, particularly for cervical pathologies such as myelopathy and radiculopathy, requires a blend of theoretical knowledge and practical skill. The complexity of these conditions, often necessitating surgical intervention, underscores the need for intricate understanding and precision in execution. Advancements in neurosurgical training, especially with the use of low-cost 3D models for simulating cervical spine tumor removal, are revolutionizing this field. These models provide the realistic and hands-on experience crucial for mastering complex neurosurgical techniques, filling gaps left by traditional educational methods. Materials and Methods: This study aimed to assess the effectiveness of 3D-printed cervical vertebrae models in enhancing surgical skills, focusing on tumor removal, and involving 20 young neurosurgery residents. These models, featuring silicone materials to simulate the spinal cord and tumor tissues, provided a realistic training experience. The training protocol included a laminectomy, dural incision, and tumor resection, using a range of microsurgical tools, focusing on steps usually performed by senior surgeons. Results: The training program received high satisfaction rates, with 85% of participants extremely satisfied and 15% satisfied. The 3D models were deemed very realistic by 85% of participants, effectively replicating real-life scenarios. A total of 80% found that the simulated pathologies were varied and accurate, and 90% appreciated the models’ accurate tactile feedback. The training was extremely useful for 85% of the participants in developing surgical skills, with significant post-training confidence boosts and a strong willingness to recommend the program to peers. Conclusions: Continuing laboratory training for residents is crucial. Our model offers essential, accessible training for all hospitals, regardless of their resources, promising improved surgical quality and patient outcomes across various pathologies.

## 1. Introduction

The intricate domain of spinal surgery necessitates a comprehensive amalgamation of theoretical understanding, practical proficiency, and profound insight into spinal pathologies [1]. Cervical pathology, encompassing conditions such as cervical myelopathy and radiculopathy, often mandates surgical intervention to restore neural function and alleviate discomfort [2]. The pathophysiology of these conditions mirrors the complexity of the anatomical structures they affect, presenting symptoms like pain, numbness, and functional impairment [3,4].

In the neurologic, oncologic, mechanical, and systemic (NOMS) decision framework, neurologic considerations, particularly epidural spinal cord compression and functional radiculopathy, assume paramount importance [5]. Tumors, whether intradural-intramedullary, intradural-extramedullary, or extradural, emerge as significant etiologies of spinal cord and root compression [5]. Schwannomas, predominantly intradural extramedullary spinal tumors, are frequently linked to neurofibromatosis type 2 [6]. Spinal cord ependymomas predominantly manifest intradurally, constituting a quarter of intramedullary spinal cord tumors [7]. Spinal metastases, affecting one-fifth of cancer patients, commonly precipitate spinal cord compression [5].

In an era of technological progress, 3D models have emerged as potent tools for augmenting neurosurgical training [8,9,10]. They furnish detailed spatial awareness essential for surgical strategizing and execution, alongside facilitating practical simulations [11]. The cost-effectiveness and adaptability of 3D-printed models democratize access to premium training materials [12]. Immediate visual and tactile feedback enriches learning experiences, fostering active engagement and collaborative learning environments [13,14,15,16]. As medical knowledge evolves, the versatility of 3D models ensures their continued relevance in neurosurgical education [17]. Their deployment curtails reliance on cadaveric specimens and live animals, aligning educational methodologies with ethical standards [18]. The integration of 3D models into curricula instills a sense of professional responsibility and ethical conduct among residents, fortifying ethical practice standards.

## 2. Materials and Methods

### 2.1. Study Design

This prospective educational intervention study was conducted at the Federal Center of Neurosurgery, Tyumen, Russian Federation. The study’s primary objective was to evaluate the efficacy of 3D-printed models of cervical vertebrae in enhancing the participants’ learning experience and surgical skills, emphasizing the steps of tumor removal. A total of 20 participants (neurosurgery residents), were enrolled and included in this study.

### 2.2. 3D Model Creation

Utilizing DICOM files from patient CT scans, precise anatomical models of the cervical spine were generated, spanning from the atlas (C1) to the cervicothoracic junction. These models were created using Horos^®^ 4.0 [19], a freely available software, which employs Hounsfield Unit thresholds to construct 3D vectorial representations, prioritizing cortical bone with its elevated density. While the resulting mesh exhibits high fidelity, potential artifacts were addressed through refinement using Meshmixer^®^ 3.5 [20] from Autodesk (Figure 1). The finalized 3D model was then imported into Cu-ra^®^ 4.6 software [21] for parameter configuration and exported in “gcode” format, enabling the Anet A8^®^ 3D printer to produce the spinal replicas with the fused material. The selected material was chosen for its resilient, yet slightly less dense properties compared to natural bone, ensuring suitability for simulation purposes without compromising durability.

Within the vertebral canal, a cost-effective, manually shaped silicone material was employed to simulate the spinal cord. To represent intradural tumors, compressed gauze pieces coated with an adhesive fabric were inserted at various levels, totaling three tumors per model. Surrounding the cervical canal, a handcrafted silicone tube dyed with a specific type of non-toxic dye was utilized to imitate the dura mater. The dorsal aspect of this tube was lined with a layer of synthetic material made from simple cardboard to simulate the ligamentum flavum (Figure 2). Silicone tubes, stained with a reddish dye to mimic blood vessels, were strategically positioned along the vertebral foramina from C6 to C1 (Figure 3). Additionally, small manually cut white silicone tubes were used, extending from inside the spinal cord to the dura mater and passing through and out of the spinal canal through the conjunction foramina. Furthermore, a basic knotted string was placed between the spinous processes to represent the interspinous ligaments. An infusion system was connected and fixed with forceps at the cephalic border of the model to continuously irrigate water inside the dura mater.

### 2.3. Simulation Training Protocol

The training involved 20 first- and second-year neurosurgery residents, using standard surgical equipment. Simulation included laminectomy, ligamentum flavum removal, dural incision, and tumor resection with continuous CSF flow simulation. The simulation ended with dural closure and simulated laminoplasty. Initial surgical approaches were excluded, focusing on emulating the surgical steps, from laminectomy to dural closure, typically performed by senior neurosurgeons. This strategy allows residents to practice intermediate steps in a controlled environment before performing the full surgery in the operating room (Figure 4). Notably, Figure 5 illustrates the supplementary training provided in the laboratory setting for intermediate steps.

### 2.4. Simulation Fidelity

To augment the fidelity of the simulation, 3D silicone models were specifically crafted based on the low-cost model mentioned above. By measuring the dimensions of the human spinal cord and cervical canal, varying densities and types of silicone were employed to accurately replicate the physical properties of different spinal tissues. A soft and pliable silicone (Eco-flex™ 00-30, Smooth-On, Chicago, IL, USA) was utilized to mimic the delicate consistency of the spinal cord, providing realistic feedback during manipulations. For the simulation of tumor tissue, a firmer silicone (Dragon Skin™ 10 Medium, Smooth-On, Chicago, IL, USA) was chosen, with its texture modified by adding silicone thickeners or encapsulating materials, such as silicone-based non-Newtonian fluids, to simulate the variable resistance of neoplastic tissues. Additionally, to recreate the feel of tougher structures, such as the calcified regions within tumors or ossified ligaments, a more rigid silicone (Smooth-Cast™ 300, Smooth-On, Chicago, IL, USA) was selectively employed. This strategic use of different silicone types, with varying Shore hardness scales and textural characteristics, imbues the models with a multisensory dimension that closely parallels the experience of actual surgical scenarios. The tactile nuances provided by this array of materials are instrumental in developing the dexterity and sensory perception required for the complex task of spinal tumor resection (Figure 1).

### 2.5. Quality Control

#### 2.5.1. Pre-Processing Verification, 3D Printer Calibration and Tumor Placement Standardization

Before processing for 3D modeling, each CT dataset underwent thorough verification to ensure completeness and accuracy. Any discrepancies or artifacts within the data were corrected or noted to prevent inaccuracies in the final model. Our 3D printers underwent calibration before each printing session, including checking the print bed level, ensuring optimal nozzle clearance, and adjusting print speed and temperature settings to match the material specifications. A calibration print was run at the start of each session to ensure the output met the required tolerances. The placement of tumor simulants within the models was standardized using specially designed jigs/molds (small pieces of gauze inside tape) of several sizes (volume 1–2 × 1 cm) with long biopsy forceps and endoscopy endonasal forceps. This ensured the consistent positioning of tumors in their anatomically correct locations and cervical levels across all models, replicating the diversity of scenarios encountered in clinical practice (Figure 6).

#### 2.5.2. Material Mixing, Curing, Anatomical Fidelity, and Dimensional Assessment

The process of mixing and curing silicone materials was standardized to achieve the correct densities and consistencies needed to represent different tissue types. This involved precise measurements and mixing ratios, vacuum degassing to remove air bubbles, and controlled curing conditions to ensure the material properties were consistent with human tissue. After production, each model underwent a thorough post-production inspection to assess anatomical accuracy and fidelity to the CT-derived specifications, including dimensional assessment using digital calipers and comparison against the original CT data. This step included visual inspection by trained anatomists, as well as physical verification by expert neurosurgeons to ensure that the models provided an authentic representation of the surgical field. The texture and hardness of the silicone materials were also evaluated using tactile feedback from neurosurgical consultants to ensure that they provided a realistic simulation of tissue resistance. Models were subjected to a series of handling tests to ensure their durability and to assess how well they stood up to repeated use (3 participants worked at different levels in each model), including simulating surgical manipulations and instrument interactions.

### 2.6. Ethical Material Selection, Waste Reduction, and Sustainable Procurement

Recognizing the environmental impact of medical model production, our team proactively selected materials aligned with sustainability goals. Biodegradable silicones and recyclable supporting materials were utilized whenever possible to minimize our environmental footprint. Engaging with suppliers, we sourced eco-friendly alternatives and adopted a lifecycle perspective in material choices, considering the environmental impact from production to disposal. Precision printing techniques were employed to minimize excess material use, and recycling protocols were implemented for unavoidable waste. Design and production processes were optimized to maximize material efficiency, reducing waste generation.

### 2.7. Post-Training Evaluation and Statistical Analysis

Following the training sessions, residents assessed the efficacy of the training using newly introduced models. The evaluation process was conducted through a survey (see Appendix A). Statistical analysis was carried out using SPSS software (version 26.0, Statistical Package for the Social Sciences, IBM Inc., New York, NY, USA). Frequencies and percentages of resident responses were determined, followed by chi-square testing to ascertain whether there were statistically significant differences among the responses.

## 3. Results

### 3.1. Participant Demographics

The data indicate a significant gender disparity among respondents, with a notably higher proportion of males (90.0%) compared to females (10.0%), as confirmed by the chi-square test (χ² = 16.4, *p* < 0.001). However, no significant difference existed between first-year (50.0%) and second-year (50.0%) residents (*p* > 0.05).

### 3.2. Realism of 3D Models, Accuracy of Pathologies Simulated, Tactile Feedback and Usefulness for Surgical Skill Development

Regarding overall satisfaction, a significant majority of respondents (85.0%) reported being extremely satisfied, with only a minority expressing lower levels of satisfaction (χ² = 74.7, *p* < 0.001) (Figure 7a). The realism of the 3D models was perceived favorably, with 85.0% of respondents rating them as very realistic (χ² = 74.7, *p* < 0.001) (Figure 7b).

The variety and accuracy of simulated pathologies were highly praised, with 80.0% of respondents acknowledging them as highly varied and accurate (χ² = 46.5, *p* < 0.001). Additionally, tactile feedback from the models received widespread acclaim, with 90.0% of respondents finding the tactility to be very accurate (χ² = 61.5, *p* < 0.001) (Figure 8).

The usefulness of 3D models for surgical skill development was highly recognized, with 85.0% of respondents rating them as extremely useful (*p* < 0.001). Furthermore, the quality of the supporting materials was predominantly rated as excellent (70.0%) (*p* < 0.001) (Figure 9).

### 3.3. Integration with Overall Learning Experience and Skill Enhancement Areas

The model training demonstrated a significant integration with participants’ overall learning experiences, with a majority reporting positive outcomes. Specifically, 80.0% of respondents found that the training complemented their existing knowledge well, indicating a harmonious relationship between the training content and their prior understanding. Moreover, 70.0% stated that the training filled gaps in their practical training, suggesting that utilization of the models effectively addressed deficiencies in hands-on experience. Additionally, 85.0% noted that the training offered a new perspective on surgical techniques, highlighting its potential to broaden participants’ understanding beyond conventional methods. In terms of skill enhancement, participants reported notable improvements across various domains. For instance, 85.0% noted enhanced tumor removal accuracy, underscoring the effectiveness of the training in refining surgical precision. Furthermore, 65.0% highlighted improvements in time-management skills, while 85.0% reported enhancements in surgical technique finesse. However, only 20.0% noted better application of anatomical knowledge, and 40.0% emphasized improvements in teamwork and communication, suggesting varied impacts on different skill sets. Feedback on quality-control aspects revealed generally positive perceptions, with a majority of respondents acknowledging the consistency of high-quality models (85.0%) and accurate placement of tumors within the models (70.0%). However, 15.0% noted inconsistencies in model quality, suggesting areas for refinement (Figure 10).

### 3.4. Post-Training Confidence and Willingness to Recommend It

Using a post-training evaluation survey for evaluation, the confidence in handling real cases post training significantly increased for 80% of participants, with an additional 20% reporting a moderate increase (*p* < 0.001). This is a strong indicator of the training’s effectiveness in boosting clinical confidence (Figure 11). A high willingness to recommend the training to peers was observed among the participants, with 75% selecting “Highly recommend” and 25% selecting “Recommend”, indicative of the program’s perceived value among participants.

## 4. Discussion

### 4.1. Model Components and Assembly

The innovative assembly of our 3D models, utilizing various affordable materials to simulate different anatomical structures, is a key aspect of the training’s success. The use of silicone in varying densities and textures to mimic spinal cord and tumor tissues offers a realistic tactile experience, crucial for developing fine motor skills in neurosurgery [22]. The incorporation of elements like the ligamentum flavum using cardboard and the interspinous ligaments with string, although simplistic, effectively demonstrates the potential of low-cost materials in high-fidelity medical simulations. This approach not only aids in the understanding of complex spinal anatomy but also provides a safe, repeatable environment for skill enhancement [23,24,25].

One of the primary challenges in the 3D printing process is ensuring precise print bed calibration. An uneven print bed can lead to print failures, warping, and inaccuracies in the final model. To address this, we implemented a rigorous calibration protocol before each printing session. This included manual leveling, where the bed was adjusted manually to achieve a uniform height across the entire surface, and the use of automated bed leveling sensors that detected and corrected any discrepancies in real time. Routine maintenance, including regular cleaning and checking of the print bed, was also conducted to maintain its optimal condition. Achieving the correct densities and consistencies of silicone to replicate different spinal tissues posed another significant challenge.

The properties of the materials needed to mimic the tactile feedback of real tissues accurately. To ensure this, we used precise mixing ratios to ensure consistency and employed vacuum degassing to remove air bubbles from the silicone, resulting in smoother and more uniform materials. Controlled curing conditions were maintained to ensure that the silicone cured under consistent temperatures and humidity levels, achieving the desired hardness and flexibility. Ensuring that the models could withstand repeated use without significant wear and tear was critical for the training program’s success. To address this, we selected high-quality materials that could endure extensive handling and multiple simulation sessions.

### 4.2. Complexities of Cervical Spine Tumors

Replicating the intricate details of cervical spine anatomy accurately was a complex task. We faced challenges in modeling small and delicate structures such as nerve roots and blood vessels. Our solutions included using high-resolution printing capable of capturing fine details, manually refining printed models using tools like Meshmixer^®^ to correct any inaccuracies, and adopting a layer-by-layer approach, printing the models in layers, and assembling them to ensure that each component was as accurate as possible.

Cervical spine tumors, whether primary or metastatic, present unique challenges due to the complexity of the cervical anatomy and the critical neurovascular structures in proximity. The cervical spine is a highly mobile region with a dense concentration of essential neural structures, making tumor resection procedures particularly delicate and demanding. The diversity in the types of tumors, ranging from benign lesions like schwannomas, to aggressive malignancies such as metastases, adds to the complexity of surgical management.

### 4.3. Surgical Considerations

The surgical approach to cervical spine tumors varies significantly based on the tumor’s location, size, and pathology. Intradural tumors, for example, require meticulous dissection to avoid damage to the spinal cord and nerve roots. Extradural tumors may involve vertebral bodies and necessitate spinal stabilization procedures post resection. The surgeon must balance tumor removal with the preservation of spinal stability and neurological function, a task that demands a high degree of skill and precision.

### 4.4. Role of 3D Models in Understanding Tumor Dynamics

3D models are invaluable in providing realistic representations of these diverse tumor scenarios. They allow surgeons to appreciate the three-dimensional relationship between the tumor and surrounding structures, which is crucial in planning the surgical approach and in anticipating potential challenges. For instance, 3D models can help to visualize the extent of bone involvement by extradural tumors or the relationships of intradural tumors with the spinal cord.

### 4.5. Simulation Training Protocol, Simulation Fidelity and Quality Control

The structured training protocol, focusing on the critical aspects of spinal surgery, such as laminectomy, dural incision, and tumor resection, was tailored to bridge the gap between theoretical knowledge and practical expertise. Excluding the initial stages of surgery, such as skin incision and muscle dissection, enabled residents to concentrate on the advanced surgical steps typically performed by more experienced surgeons. This strategic focus is reflected in the survey results, where a significant majority of residents found the training extremely useful for their skill development.

The strategic use of different types of silicone to replicate the varying consistencies of spinal tissues is a testament to the model’s fidelity. The ability to provide residents with a realistic feel of different tissues underpins the efficacy of these models in surgical training. The positive feedback on the tactile accuracy of the models underscores their effectiveness in mimicking real-life surgical scenarios, thereby enhancing the residents’ manual dexterity and sensory perception.

Our stringent quality-control measures, from the verification of CT data to the post-production inspection of models, ensured high fidelity and anatomical accuracy. The attention to detail in every step of the model creation process reflects our commitment to providing the highest quality training tools.

Expert feedback was solicited from experienced neurosurgeons who evaluated the anatomical realism and tactile feedback of the models. The models underwent iterative refinement based on feedback to better replicate the surgical environment. Incorporating expert neurosurgeon feedback into the manuscript strengthens the validation of the models’ anatomical accuracy and enhances the study’s credibility.

The fact that most of participants found the models to be of high quality and appreciated the accurate tumor placement validates our quality-control efforts.

### 4.6. Ethical Considerations, Sustainable Sourcing and Survey Findings

Our ethical approach to sourcing patient data, adhering to HIPAA regulations, and ensuring patient consent, along with our focus on using sustainable and environmentally friendly materials, sets a precedent in the ethical development of medical training aids. This conscientious approach not only aligns with modern ethical standards but also imparts a sense of responsibility towards patient privacy and environmental stewardship among the residents.

The overwhelmingly positive response from the residents, as reflected in the survey results, validates the effectiveness of the 3D-model training. The high levels of satisfaction, perceived realism of the models, and the reported increase in post-training confidence are strong indicators of the program’s success. However, areas such as anatomical knowledge application and teamwork, which received relatively lower scores, highlight potential areas for improvement in future iterations of the training program.

During the COVID-19 pandemic, cadaveric practices were significantly reduced, demonstrating that, in the face of adversity, human specimen practice can be significantly impacted [26,27]. Simulation with 3D printing allows for the creation of several models from one patient and for the process to be reproduced as many times as needed for training. While 3D printing has many advantages, this process has some limitations [28]. The materials available are limited by their thermodynamic characteristics. For example, complex and large-scale models may result in deficient printing, as they are time-consuming and difficult to print [29]. Rapid prototyping allows for the quick manufacture of 3D models from medical imaging data, giving the surgeon the possibility of visual 2D information and tactile feedback from the 3D model [30]. It also enables training in order to plan the best surgical approach for the surgery.

### 4.7. Integration in Medical Education and Practice

There is a resounding consensus on the need to integrate 3D-printing technology more extensively in medical education and practice, with 100% of the respondents likely to use 3D models in their future practice. This unanimous acknowledgment emphasizes the evolving role of 3D models as indispensable tools in modern medical education and clinical practice, shaping the future trajectory of medical training and patient care methodologies [31,32,33,34].

Fused filament fabrication (FFF) 3D printing is available and widespread. PLA (polylactic acid) has many advantages, such as non-toxicity, and it is made by lactic acid or lactide polymerization. It is, therefore, not an oil derivative but is produced by the bacterial fermentation of carbohydrates (corn, carp, cassava). The sterilization process is carried out with low-temperature sterilization, and 100% ethylene oxide is the standard protocol since it does not affect the physical or anatomical properties of the material [35,36]. At the same time, the use of PLA has its drawbacks since it has a low temperature of crystallization (55 °C) and melting (180 °C). When drilling, it is important to use a low-speed drill to prevent melting [37,38].

### 4.8. Enhanced Precision and Safety in Surgical Training

One of the paramount benefits of employing 3D models in cervical spine tumor training is the significant enhancement of precision and safety [39]. These models, meticulously crafted to mirror the complex anatomy of the human spine, provide a realistic platform for residents to practice the delicate maneuvers required in tumor resection. [40]. The intricacies of spinal-cord and nerve-root handling, critical in avoiding intraoperative complications, can be safely rehearsed [38,41]. This hands-on experience is instrumental in cultivating a deeper understanding of the spatial relationships between the tumor and critical neurovascular structures, thus reducing potential risks during actual surgeries [42,43,44].

### 4.9. Bridging the Gap between Theoretical Knowledge and Clinical Application

The 3D models serve as a bridge between theoretical knowledge and clinical application. By simulating a variety of tumor types and locations within the cervical spine, residents are exposed to a wide range of scenarios, each presenting unique challenges and learning opportunities [45,46]. This exposure is invaluable in a field where experience and familiarity with diverse pathologies directly correlate with surgical proficiency and patient outcomes [47,48].

### 4.10. Customization and Personalization of Training

A unique advantage of 3D modeling is the ability to customize models based on actual patient cases [49]. This personalization allows residents to engage in patient-specific rehearsal, a concept that is gaining traction in modern surgical training [50,51]. Such personalized simulations enable trainees to pre-plan surgical strategies and anticipate challenges, thereby increasing the success rate and reducing operative times in actual surgeries [52,53,54].

### 4.11. Assessment, Feedback, Scalability and Accessibility

The utilization of 3D models also facilitates a more structured assessment of surgical skills. Instructors can provide immediate feedback on technique, decision making and problem-solving skills in a controlled environment [55,56]. This real-time feedback is crucial for the iterative learning process, allowing residents to make changes and improvements to their surgical approach and technique [57].

The scalability and accessibility of 3D printed models cannot be overstated. These models can be produced at a relatively low cost and distributed widely, making advanced surgical training more accessible to institutions with limited resources [58,59]. This democratization of surgical education is pivotal in elevating the overall standard of care, especially in regions with scarce access to high-quality training materials [60,61,62].

### 4.12. Future Directions

Looking ahead, the integration of virtual reality (VR) and augmented reality (AR) with 3D printed models presents a promising avenue for further enhancing neurosurgical training [63,64]. Combining the tactile feedback of physical models with the immersive and interactive experience of VR/AR technologies could revolutionize the way surgical skills are acquired and refined [65,66,67].

### 4.13. Limitations of This Study

The study acknowledges the inherent limitations in the 3D-printing processes used to create anatomical models. These limitations can impact the models’ quality, accuracy, and utility, potentially affecting the learning experience and outcomes for the participants. This section presents the detailed points regarding the printing limitations. The models were created using Polylactic Acid (PLA), a material that may not accurately replicate biological tissues’ properties, feel, and texture, affecting the realism and learning experience.

We are also exploring the use of resin materials, which have a consistency more accurate to real bone. Resin-based 3D printing can provide more accurate anatomical details and better simulate the hardness and feel of bone.

However, the cost of resin materials and the availability of photopolymer 3D-printing machines are significant considerations. Resin materials and the necessary photopolymer printers are more expensive and less widely available compared to PLA and FFF (Fused Filament Fabrication) printers. This limits their use to institutions with higher budgets and access to advanced printing technology.

The availability of materials is constrained by their thermodynamic characteristics, which might limit the range and types of models that can be created.

Any imperfections in the printing process can affect the anatomical fidelity of the models. Ensuring high accuracy requires rigorous quality-control measures, including pre-processing verification, printer calibration, and post-processing refinement.

Despite generating high-fidelity meshes, potential artifacts need to be addressed through meticulous refinement to maintain model accuracy and educational value.

The current study relies on subjective evaluations by participants to measure effectiveness. Developing comprehensive validation frameworks that include objective metrics, standardized assessment tools, and longitudinal studies is essential to robustly measure the improvement in surgical skills.

Incorporating control groups undergoing traditional training methods would provide a more robust comparison and strengthen the evidence supporting the effectiveness of the 3D model-based training.

## 5. Conclusions

It is essential to continue to promote the training of residents in the laboratory before performing any surgery, in this case posterior approaches to the cervical spine. If not, it is still indispensable to combine laboratory practice with participation in surgery, and not only to focus on pure “old school” or low-resource neurosurgery with trial and error during the surgical act on the patient without having previously undergone quality training, as this also represents a serious ethical problem. Therefore, we conclude that our model provides indispensable help to residents to complement their training and is accessible to low-income hospitals or large hospitals. The model will undoubtedly improve the quality of surgeries in the future in our guild, with a consequent improvement in the care and outcome of patients regardless of the pathology.

## Figures and Tables

**Figure 1 brainsci-14-00547-f001:**
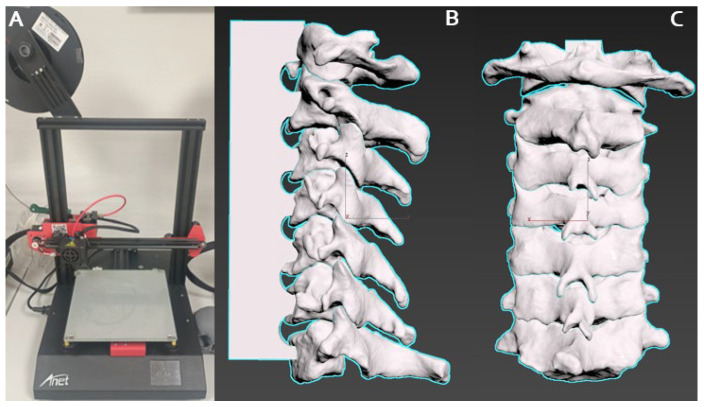
Cervical spine 3D model creation with FFF 3D printer (Anet A8^®^, Anet Technology Co., Shenzhen, China): (**A**) high-resolution cervical spine anatomical models spanning from the atlas and axis to the cervicothoracic junction taken DICOM files from CT scan and analyzed with Horos (free software) for the creation of a 3D vectorial model; (**B**) lateral view; and (**C**) dorsal view.

**Figure 2 brainsci-14-00547-f002:**
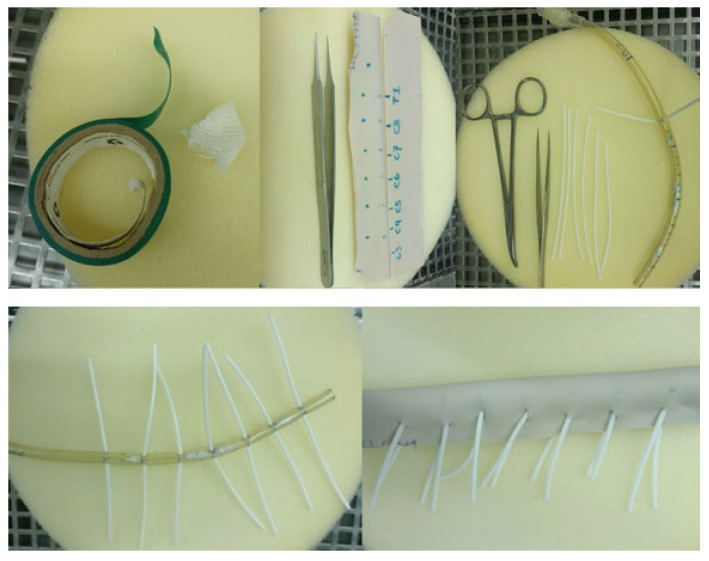
Cost-effective and manually shaped silicone material and endotracheal tube to simulate the spinal cord, dura mater and spinal roots, as also the surgical and microsurgical instruments set material used during the training simulation.

**Figure 3 brainsci-14-00547-f003:**
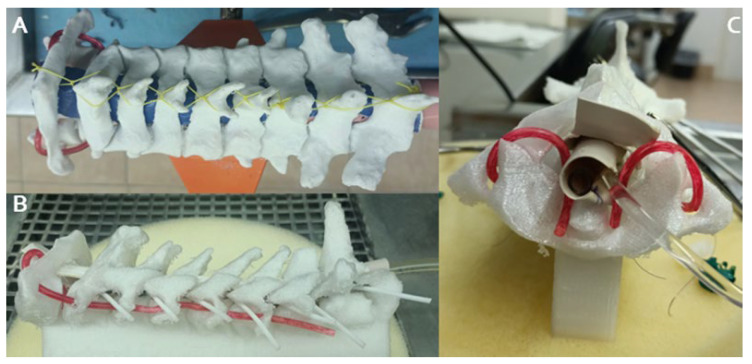
Dorsal (**A**); lateral (**B**); and axial proximal (**C**) views of the final 3D model with all cost-effective components include inside of the spinal canal: Silicon spinal cord, silicon dura mater, ligaments, vertebral arteries, roots through the conjunction foramen and irrigation water system in the cephalic border.

**Figure 4 brainsci-14-00547-f004:**
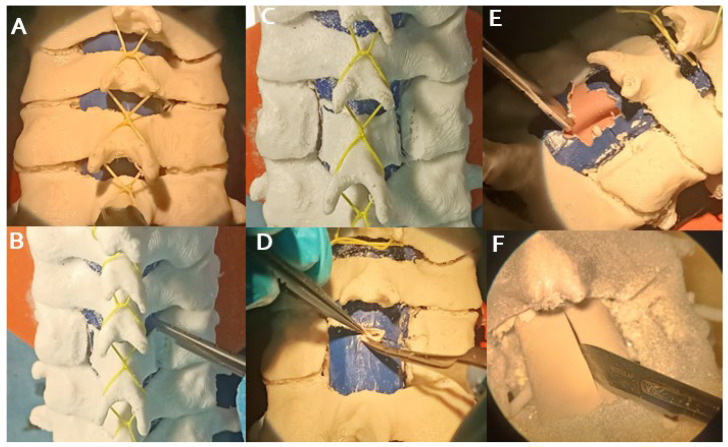
Sequence of steps at the beginning of the simulation of the surgery, from exposure and laminectomy (**A**–**C**); opening and removal of the yellow ligament (**D**–**F**); and dural opening with linear incision in the midline.

**Figure 5 brainsci-14-00547-f005:**
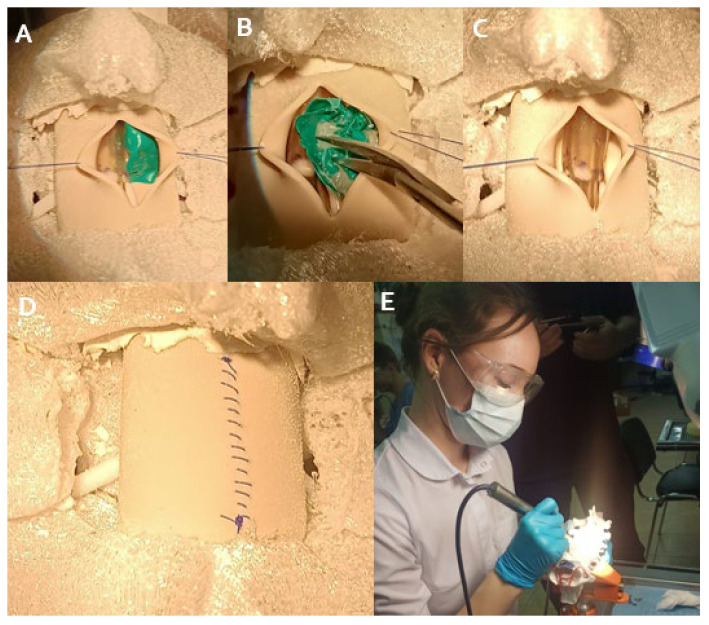
Sequence of steps of the simulation, exposition of the intradural extramedullary tumor (**A**); dissection, debulking and removal of the tumor (**B**,**C**); closing the dura mater (**D**); and a resident at work with the 3D model, with the drill and other instruments of the laboratory (**E**).

**Figure 6 brainsci-14-00547-f006:**
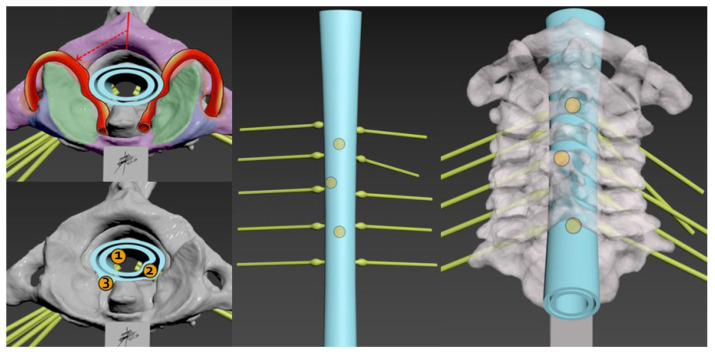
Verifying exactitude of anatomical structures and placement and design of tumors prototypes in their various anatomic locations; intradural intramedullary (1), intradural extramedullary (2), extradural (3) and the selection of different cervical levels intercalated to make possible the performance of the work by multiple participants (dorsal views) in a single model, thus optimizing resources to the maximum.

**Figure 7 brainsci-14-00547-f007:**
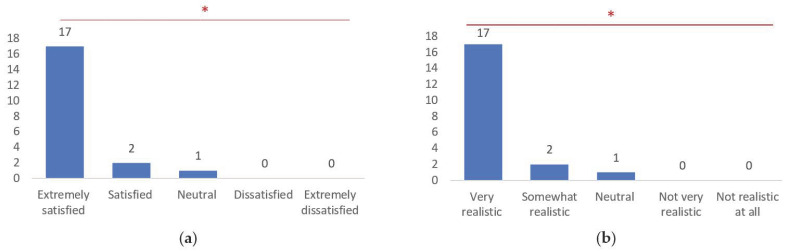
Overall satisfaction related to the utilization of 3D models for medical training (**a**); and realism of the 3D models (**b**). *, statistically significant differences recognized by chi-square test (*p* < 0.001).

**Figure 8 brainsci-14-00547-f008:**
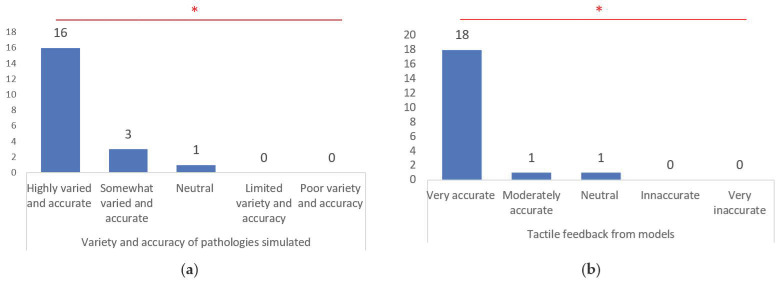
Variety and accuracy of pathologies simulated (**a**); and tactile feedback from models (**b**). *, statistically significant differences recognized by chi-square test (*p* < 0.001).

**Figure 9 brainsci-14-00547-f009:**
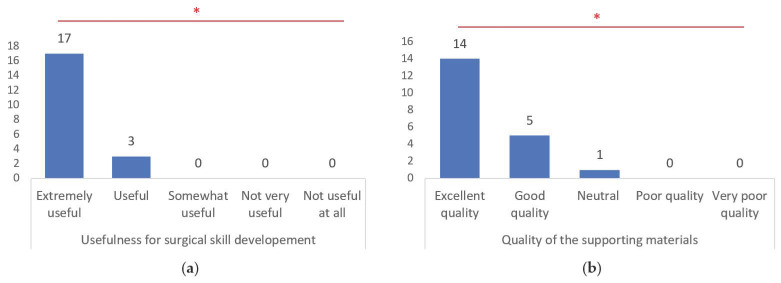
Usefulness for surgical skill development (**a**); and quality of the supporting materials (**b**). *, statistically significant differences recognized by chi-square test (*p* < 0.001).

**Figure 10 brainsci-14-00547-f010:**
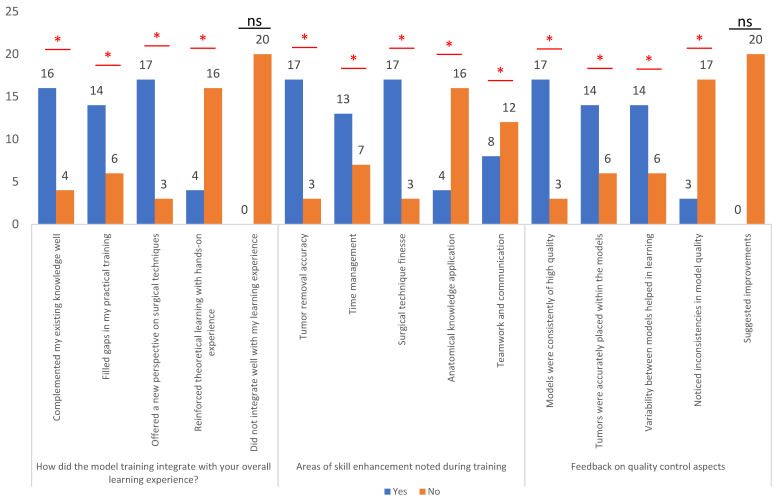
Overall learning experience, area of skill enhancement noted during training and feedback on quality-control aspects. * Represent the questions where dual response was noted (Yes and No); ns represent questions where only a single response was noted.

**Figure 11 brainsci-14-00547-f011:**
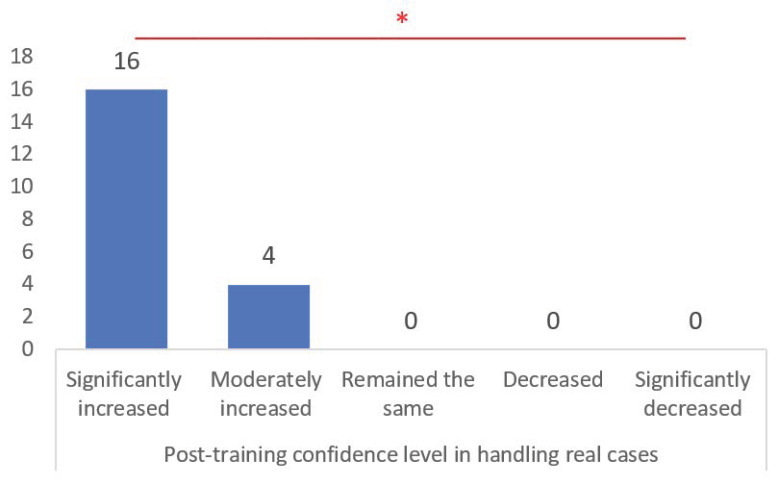
The graph shows participant ratings of experience in post-training confidence level in handling real cases. *, statistically significant differences recognized by chi-square test (*p* < 0.001).

## Data Availability

Data is contained within the article.

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
