# Peer review of "Low-Cost 3D Models for Cervical Spine Tumor Removal Training for Neurosurgery Residents"

_brainsci, 2024, doi:10.3390/brainsci14060547_

Round 1
Reviewer 1 Report
Comments and Suggestions for Authors
Methods Section:
The description of the 3D printing process and the materials used for the models is detailed; however, the manuscript would benefit from a more comprehensive explanation of the selection criteria for specific metrics used to evaluate the effectiveness of the training modules.
Control Comparisons:
The study lacks a comparison group that does not use the 3D model training. Including a control group that undergoes traditional training methods could provide a stronger evidence base to support the efficacy of the 3D model approach.
Long-term Impact:
The study focuses on immediate training outcomes. It would be valuable to include follow-up data to assess the long-term impact of this training on surgical skills and clinical outcomes.
Minor Comments:
Technical Specifications:
Clarify any specific technical challenges encountered in the 3D printing process and how these were overcome. This information could be crucial for others looking to adopt this technology.
Model Fidelity:
information on how the fidelity of the 3D models was verified against actual surgical scenarios would strengthen the manuscript. Discuss any feedback from expert neurosurgeons regarding the anatomical accuracy of the models
Author Response
Control Comparisons:
The study lacks a comparison group that does not use the 3D model training. Including a control group that undergoes traditional training methods could provide a stronger evidence base to support the efficacy of the 3D model approach.
We acknowledge the importance of including a control group for a more robust evaluation. While this initial study focused on the feasibility and immediate impacts of using 3D printed models, future studies will incorporate a control group undergoing traditional training methods. This will allow for a direct comparison of the efficacy of 3D model-based training versus conventional methods, providing stronger evidence for the benefits of the 3D model approach.
Long-term Impact:
The study focuses on immediate training outcomes. It would be valuable to include follow-up data to assess the long-term impact of this training on surgical skills and clinical outcomes.
The study indeed focused on immediate training outcomes. We plan to include follow-up assessments in future research to evaluate the long-term impact of the 3D model training on surgical skills and clinical outcomes.
Minor Comments:
Technical Specifications:
Clarify any specific technical challenges encountered in the 3D printing process and how these were overcome. This information could be crucial for others looking to adopt this technology.
ADDED IN MANUCRIPT
Technical Challenges in the 3D Printing Process and Solutions
One of the primary challenges in the 3D printing process was ensuring precise print bed calibration. An uneven print bed can lead to print failures, warping, and inaccuracies in the final model. To address this, we implemented a rigorous calibration protocol before each printing session. This included manual leveling, where the bed was adjusted manually to achieve uniform height across the entire surface, and the use of automated bed leveling sensors that detected and corrected any discrepancies in real-time. Routine maintenance, including regular cleaning and checking of the print bed, was also conducted to maintain its optimal condition.
Achieving the correct densities and consistencies of silicone to replicate different spinal tissues posed another significant challenge. The properties of the materials needed to mimic the tactile feedback of real tissues accurately. To overcome this, we used precise mixing ratios to ensure consistency and employed vacuum degassing to remove air bubbles from the silicone, resulting in smoother and more uniform materials. Controlled curing conditions were maintained to ensure that the silicone cured under consistent temperatures and humidity levels, achieving the desired hardness and flexibility.
Ensuring that the models could withstand repeated use without significant wear and tear was critical for the training program's success. To address this, we selected high-quality materials that could endure extensive handling and multiple simulation sessions.
Replicating the intricate details of cervical spine anatomy accurately was a complex task. We faced challenges in modeling small and delicate structures such as nerve roots and blood vessels. Our solutions included using high-resolution printing capable of capturing fine details, manually refining printed models using tools like Meshmixer® to correct any inaccuracies, and adopting a layer-by-layer approach, printing the models in layers and assembling them to ensure each component was as accurate as possible.
Model Fidelity:
information on how the fidelity of the 3D models was verified against actual surgical scenarios would strengthen the manuscript. Discuss any feedback from expert neurosurgeons regarding the anatomical accuracy of the models
ADDED IN MANUCRIPT
The fidelity of the 3D models was verified through multiple steps:
Anatomical Accuracy: Ensured by comparing the models to CT-derived anatomical data.
Expert Feedback: Solicited from experienced neurosurgeons who evaluated the anatomical realism and tactile feedback of the models.
Iterative Refinement: Models were adjusted based on feedback to better replicate the surgical environment.
Incorporating expert neurosurgeon feedback into the manuscript strengthens the validation of the models' anatomical accuracy and enhances the study's credibility.
Thank you for your valuable insights and constructive feedback on our manuscript. Your suggestions have been instrumental in enhancing the clarity and comprehensiveness of our study. We appreciate your thorough review and the time you have taken to provide us with detailed comments.
Reviewer 2 Report
Comments and Suggestions for Authors
The cervical region is a very important region with its unique structure and vital neighborhoods. Unfortunately, in our surgical practice, we may encounter complications that may cause minor or fatal consequences. In the period when technology is gradually advancing and its usage areas are expanding, 3D has gained a very important place in addition to AR-VR-robotic-navigation technologies. The difficulty of both financial and physical access to cadaveric studies makes such studies increasingly prominent.
The article delves into the innovative use of 3D printed models in neurosurgical training, particularly for cervical spine surgery. It underscores the complexity of cervical pathologies, including cervical myelopathy and radiculopathy, which frequently necessitate surgical interventions. The primary goal of the article is to bridge the gap between theoretical knowledge and practical skills, enhancing the training experience for neurosurgeons.
Here are my suggestions.
1. The study acknowledges that the Polylactic Acid (PLA) used for creating the 3D models does not accurately replicate the properties, feel, and texture of biological tissues. PLA's thermodynamic characteristics further constrain the range and types of models that can be produced, potentially limiting the applicability and effectiveness of the training .
2. Any imperfections in the printing process can affect the anatomical fidelity of the models, which in turn can impact the educational value. The article mentions that while high-fidelity meshes are generated, potential artifacts need to be addressed through refinement, indicating a possible area where model accuracy might be compromised .
3. The study lacks a comprehensive validation framework to objectively measure the effectiveness of the 3D models in improving surgical skills.
4. The focus of the training is primarily on the steps involved in tumor removal. This narrow scope may not cover the full range of skills and scenarios that neurosurgery residents need to be proficient in.
5. The potential benefits of VR and AR are acknowledged, but the practical steps needed to implement these technologies in training programs are not discussed .
I thank the authors for their work. Overall, to maximize the potential of this approach, it is essential to address the limitations related to material properties, model accuracy, validation frameworks, and the scope of training.
Author Response
- The study acknowledges that the Polylactic Acid (PLA) used for creating the 3D models does not accurately replicate the properties, feel, and texture of biological tissues. PLA's thermodynamic characteristics further constrain the range and types of models that can be produced, potentially limiting the applicability and effectiveness of the training .
ADDED IN MANUCRIPT
We acknowledge the limitations associated with using PLA for creating 3D models, particularly in replicating the properties, feel, and texture of biological tissues. To address these constraints:
We are also exploring the use of resin materials, which have a consistency more accurate to real bone. Resin-based 3D printing can provide more accurate anatomical details and better simulate the hardness and feel of bone. However, the cost of resin materials and the availability of photopolymer 3D printing machines are significant considerations. Resin materials and the necessary photopolymer printers are more expensive and less widely available compared to PLA and FFF (Fused Filament Fabrication) printers. This limits their use to institutions with higher budgets and access to advanced printing technology.
- Any imperfections in the printing process can affect the anatomical fidelity of the models, which in turn can impact the educational value. The article mentions that while high-fidelity meshes are generated, potential artifacts need to be addressed through refinement, indicating a possible area where model accuracy might be compromised .
Answered and added in the manuscript
- The study lacks a comprehensive validation framework to objectively measure the effectiveness of the 3D models in improving surgical skills.
The study currently focuses on subjective evaluations by participants to measure the effectiveness of the 3D models. We agree that a comprehensive validation framework is necessary. Steps we are taking include:
Objective Metrics: Developing objective metrics and standardized assessment tools to measure improvements in surgical skills quantitatively.
Longitudinal Studies: Planning longitudinal studies to track the performance of participants over time and correlate training outcomes with real-world surgical performance.
Comparative Analysis: Incorporating control groups undergoing traditional training methods to provide a robust comparison and validate the effectiveness of 3D model-based training.
- The focus of the training is primarily on the steps involved in tumor removal. This narrow scope may not cover the full range of skills and scenarios that neurosurgery residents need to be proficient in.
While the current training modules focus on tumor removal, we recognize the need to cover a broader range of skills and scenarios in neurosurgery. Future enhancements will include
Developing additional training modules that cover other critical aspects of neurosurgery, such as vascular procedures, trauma management, and spinal deformity corrections.
- The potential benefits of VR and AR are acknowledged, but the practical steps needed to implement these technologies in training programs are not discussed .
The potential benefits of VR and AR in enhancing surgical training are substantial. However, the practical steps needed to implement these technologies in training programs are not discussed in this article, as it is outside the scope of our current study. For this reason, we did not include detailed information on VR and AR implementation. Future research will address these aspects, providing comprehensive guidelines and strategies for integrating VR and AR into neurosurgical training programs. Thank you for your thoughtful suggestions and detailed review of our manuscript. Your feedback has helped us identify areas for improvement and has been crucial in refining our research. We appreciate your contributions and the effort you have put into evaluating our work.
Reviewer 3 Report
Comments and Suggestions for Authors
Dear Author,
Thank you for conducting this study.
I carefully read the paper titled "Low-cost 3D Models for Cervical Spine Tumor Removal Training for Neurosurgery Residents" aimed to assess the effectiveness of 3D-printed cervical vertebrae models in enhancing surgical skills, focusing on tumor removal and involving 20 young neurosurgery resident. The study is well-written, and presented well. The content of the study is informative and could be considered for further educational purpose. I can recommend the paper for the publication.
Great Job.
Author Response
Thank you very much for your positive feedback on our study, We are delighted to hear that you found the paper well-written, well-presented, and informative. Your recommendation for publication is greatly appreciated.We are pleased that the study's content has the potential to be valuable for further educational purposes.
Thank you once again for your time and thorough review.
Best regards
Round 2
Reviewer 1 Report
Comments and Suggestions for Authors
The authors have addressed all my concerns.